# Quality Aware Generative Adversarial Networks

**Parimala Kancharla,   Sumohana S. Channappayya**
Department of Electrical Engineering
Indian Institute of Technology Hyderabad
`{ee15m17p100001, sumohana}@iith.ac.in`

## Abstract

Generative Adversarial Networks (GANs) have become a very popular tool for implicitly learning high-dimensional probability distributions. Several improvements have been made to the original GAN formulation to address some of its shortcomings like mode collapse, convergence issues, entanglement, poor visual quality etc. While a significant effort has been directed towards improving the visual quality of images generated by GANs, it is rather surprising that objective image quality metrics have neither been employed as cost functions nor as regularizers in GAN objective functions. In this work, we show how a distance metric that is a variant of the Structural SIMilarity (SSIM) index (a popular full-reference image quality assessment algorithm), and a novel quality aware discriminator gradient penalty function that is inspired by the Natural Image Quality Evaluator (NIQE, a popular no-reference image quality assessment algorithm) can each be used as excellent regularizers for GAN objective functions. Specifically, we demonstrate state-of-the-art performance using the Wasserstein GAN gradient penalty (WGAN-GP) framework over CIFAR-10, STL10 and CelebA datasets.

## 1   Introduction

Generative Adversarial Networks (GANs) [Goo+14] have become a very popular tool for implicitly learning high-dimensional probability distributions. A large number of very interesting and useful applications have emerged due to the ability of GANs to learn complex real-world distributions. Some of these include image translation [Iso+17], image super-resolution [Led+17], image saliency detection [Pan+17] etc. While GANs have indeed become very popular, they suffer from drawbacks such as mode collapse, convergence issues, entanglement, poor visual quality etc. A significant amount of research effort has focused on addressing these shortcomings in the original GAN formulation. While the literature does consider the quality of the generated images as a performance metric, it is rather surprising that the use of objective image quality metrics in the GAN cost function has been very limited. We address this lacuna with our contributions as summarized below:

- We make explicit use of objective image quality assessment (IQA) metrics and their variants for regularizing WGAN with gradient penalty (WGAN-GP), and propose Quality Aware GANs (QAGANs).

- We propose a novel quality aware discriminator gradient penalty function based on the local statistical signature of natural images as in NIQE [MSB13].

- We demonstrate state-of-the-art performance on CIFAR-10, STL 10 and CelebA datasets for non-progressive GANs.

## 2 Background

We review relevant works on GANs and IQA algorithms to setup the background necessary to present our proposed quality aware GANs.

### 2.1 Generative Adversarial Networks (GANs)

Given the explosive growth in the literature on GANs, we only review representative works in the following. GANs [Goo+14] pose the the problem of learning a high-dimensional distribution from data samples in a game-theoretic framework. A typical GAN architecture consists of a *generator* modelled by a neural network and denoted by $G$ and parameterized by $\theta_g$, and a *discriminator* also modelled by a neural network denoted by $D$ and parameterized by $\theta_d$. The goal of the generator is to generate samples denoted $G(z)$ (where $z$ is a noise random variable with prior $P_z$) that "mimic" true data samples $x$ drawn from a distribution $P_r$. The discriminator's goal is to maximize its ability to tell $G(z)$ apart from $x$. The generator and discriminator engage in an adversarial combat or game to train each other (simultaneously). This game is formulated as

$$\min_G \max_D V(D, G) = E_{x \sim P_r}[\log(D(x))] + E_{z \sim P_z}[\log(1 - D(G(z)))] \tag{1}$$

The value function $V(D, G)$ is defined so that the discriminator tries to maximize the probability of assigning the correct label to $G(z)$ and $x$ via the term $E_{x \sim P_r}[\log(D(x))]$ while the generator simultaneously tries to minimize the term $E_{z \sim P_z}[\log(1 - D(G(z)))]$. The model parameters $\theta_g, \theta_d$ are learnt by solving this optimization in an iterative fashion. This formulation suffered from the drawbacks mentioned earlier - mode collapse, convergence issues while training, poor visual quality etc.

#### 2.1.1 Wasserstein GAN with Gradient Penalty (WGAN-GP)

Wasserstein GAN (WGAN) [ACB17] was one of the first works to address training issues in the original GAN formulation. It introduced the Wasserstein distance to compare distributions and showed that it possesses a number of useful convergence and continuity properties. It also showed that 1-Lipschitz functions can be used in practise to find the Wasserstein distance between distributions (at the discriminator). These 1-Lipschitz functions are realized using a neural network and the Lipschitz condition is enforced by clipping the network weights. Despite these improvements, Gulrajani et al. [Gul+17] showed that weight clipping in WGAN led to undesirable behaviour such as poor samples or failing to converge. They propose a stabler solution called WGAN with Gradient Penalty (WGAN-GP) where they penalize the norm of the discriminator's gradient with respect to its input. They showed that the norm of the discriminator gradient is in fact 1 almost everywhere for 1-Lipschitz functions. We will make use of the WGAN-GP in our work given its stability in training GANs.

#### 2.1.2 Banach Wasserstein GAN (BWGAN)

Adler and Lunz [AL18] introduced Banach WGANs (BWGANs) that provide a framework for using arbitrary norms to train Wasserstein GANs (WGANs). They note that WGANs and their variants only consider $l_2$ as the underlying metric. Specifically, they generalize the WGAN-GP framework to Banach spaces and show how WGANs can be trained with arbitrary norms. The motivation for BWGANs is to allow the generator to emphasize desired image features such as edges as demonstrated using Sobolev and $L_p$ norms. The motivation for our work comes from BWGANs and is similar in spirit. They also suggest that WGAN training can be extended to a general metric space (with metric $d(X, Y)$) by using a penalty term of the form

$$E_{X \sim P_r, Y \sim P_G} \left[ \left( \frac{|D(X) - D(Y)|}{d(X, Y)} \right) - 1 \right]^2. \tag{2}$$

#### 2.1.3 Zero-Centered Gradient Penalty Approaches

Other recent approaches to improving the stability of GAN training revisit the original formulation in [Goo+14]. Roth et al. [Rot+17] present a regularization approach where the inputs to the discriminator are smoothed by convolving them with noise (realized by adding noise to the samples during training). This is shown to result in a zero-centered gradient penalty regularizer. Mescheder et al. [MGN18]

prove that zero-centered gradient penalties make training more stable. Thanh-Tung et al. [TTV19] propose another variant of the zero-centered gradient penalty for improving the convergence and generalizing capability of GANs. These works are significant in that they provide a clear theoretical explanation to the issues in training GANs and how they can be overcome.

## 2.2 Image Quality Assessment

Objective image quality assessment (IQA) metrics can be classified into three classes depending on their use of the reference (or undistorted) image for quality assessment. Full-reference (FR) IQA metrics make use of the complete reference image while reduced reference (RR) IQA metrics make use of partial reference image information for quality prediction. No-reference (NR) IQA metrics on the other hand predict the quality of an image in a reference-free or stand-alone fashion. It should be noted that the IQA metrics assume that the images in question are of natural (photographic) scenes. In this work, we show how an FR and an NR IQA algorithm can each be used for the quality aware design of GANs.

### 2.2.1 The Structural SIMilarity (SSIM) index

It would not be an exaggeration to claim that the invention of the SSIM index [Wan+04] heralded a revolution in the design of objective quality assessment algorithms. The SSIM index is an FR IQA metric that is based on the premise that distortions lead to change in local image structure, and that the human visual system is sensitive to these structural changes. The SSIM index quantifies the change in structural information in the test image relative to the reference and computes a quality score. Specifically, the SSIM index computes changes to local mean, local variance and local structure (or correlation) and pools them to find the local quality score. These local scores are then averaged across the image to find the image quality score. This is summarized as follows.

$$\text{SSIM}(P_{(i,j)}, T_{(i,j)}) = L(P_{(i,j)}, T_{(i,j)}).C(P_{(i,j)}, T_{(i,j)}).S(P_{(i,j)}, T_{(i,j)}), \tag{3}$$

where $P, T$ refer to the pristine and test image respectively, the subscript $(i,j)$ is the pixel index, $L(P_{(i,j)}, T_{(i,j)})$, $C(P_{(i,j)}, T_{(i,j)})$, $S(P_{(i,j)}, T_{(i,j)})$ are the local luminance, contrast and structure scores at pixel $(i,j)$ respectively. Further,

$$L(P_{(i,j)}, T_{(i,j)}) = \frac{2\mu_P(i,j)\mu_T(i,j) + C_1}{\mu_P^2(i,j) + \mu_T^2(,j) + C_1}, C(P_{(i,j)}, T_{(i,j)}) = \frac{2\sigma_P(i,j)\sigma_T(i,j) + C_2}{\sigma_P{}^2(i,j) + \sigma_T{}^2(i,j) + C_2},$$
$$S(P_{(i,j)}, T_{(i,j)}) = \frac{\sigma_{PT}(i,j) + C_3}{\sigma_P(i,j)\sigma_T(i,j) + C_3}, \tag{4}$$

where $\mu_P(i,j) = \frac{1}{(2K+1)^2} \sum\limits_{m=-K}^{K} \sum\limits_{n=-K}^{K} P(i-m, j-n)$, $\sigma_P^2(i,j) = \frac{1}{(2K+1)^2} \sum\limits_{m=-K}^{K} \sum\limits_{n=-K}^{K} (P(i-m, j-n) - \mu_P(i,j))^2$ are the local mean and variance of the pristine image patch of size $(2K+1) \times (2K+1)$ centered at $(i,j)$. $\mu_T(i,j), \sigma_P^2(i,j)$ are defined similarly for the test image $T$. The cross covariance is defined as $\sigma_{PT}(i,j) = \frac{1}{(2K+1)^2} \sum\limits_{m=-K}^{K} \sum\limits_{n=-K}^{K} (P(i-m, j-n) - \mu_P(i,j)) \times (T(i-m, j-n) - \mu_T(i,j))$. The constants $C_1, C_2, C_3$ are used to avoid division-by-zero issues. For simplicity, $C_3 = C_2/2$ in the standard implementation which leads to

$$\text{SSIM}(P_{(i,j)}, T_{(i,j)}) = L(P_{(i,j)}, T_{(i,j)}).CS(P_{(i,j)}, T_{(i,j)}), \tag{5}$$

where

$$CS(P_{(i,j)}, T_{(i,j)}) = \frac{2\sigma_{PT}(i,j) + C_2}{\sigma_P{}^2(i,j) + \sigma_T{}^2(i,j) + C_2}. \tag{6}$$

The image level SSIM index is given by:

$$\text{SSIM}(P, T) = \frac{1}{M \times N} \sum_{i=1}^{M} \sum_{j=1}^{N} \text{SSIM}(P_{(i,j)}, T_{(i,j)}), \tag{7}$$

where the images are of size $M \times N$ (assuming appropriate boundary handling).

### 2.2.2 Natural Image Quality Estimator (NIQE)

NIQE [MSB13] is a popular NR IQA metric that is based on the statistics of mean subtracted and contrast normalized (MSCN) natural scenes. An MSCN image $\hat{I}$ is generated from an input image $I$ according to:

$$\hat{I}(i,j) = \frac{I(i,j) - \mu(i,j)}{\sigma(i,j) + 1}, \tag{8}$$

where $(i,j)$ is the pixel index and $\mu(i,j)$ and $\sigma(i,j)$ are the local mean and standard deviation computed as in the case of SSIM index (see Section 2.2.1). The constant 1 in the denominator is to prevent division-by-zero issues. NIQE relies on the following observations about the statistics of MSCN naturals scenes: a) the statistics of MSCN natural images reliably follow a Gaussian distribution [Rud94], b) the statistics of MSCN pristine and distorted images can be modeled well using a generalized Gaussian distribution (GGD) [MB10]. NIQE, as the name suggests, quantifies the naturalness of an image. To do so, it proposes a statistical model for the class of pristine natural scenes and uses the model's parameters for quality estimation. Specifically, it models MSCN pristine image coefficients using a GGD, and models the products of neighbouring MSCN coefficients using an asymmetric GGD (AGGD). The parameters of these GGD and AGGD models are in turn modeled using a Multivariate Gaussian (MVG) distribution whose parameters $\mu_P, \Sigma_P$ are then used as representatives of the *entire class of pristine natural images*. The quality of a test image is measured in terms of the "distance" of its MVG parameters $\mu_T, \Sigma_T$ from the pristine MVG parameters. This is quantified as:

$$D(\mu_P, \mu_T, \Sigma_P, \Sigma_T) = \sqrt{(\mu_P - \mu_T)^T \left(\frac{\Sigma_P + \Sigma_T}{2}\right)^{-1} (\mu_P - \mu_T)}, \tag{9}$$

assuming that the sum of the matrices is invertible.

## 3 Quality Aware GANs (QAGANs)

While significant progress has been made in the design of objective IQA metrics, their usage as cost functions has been limited by their unwieldy mathematical formulation. Non-convexity, difficulty in gradient computation, not satisfying the properties of distances and norms are some of the impediments to their usage in formulating optimization problems. In the following, we identify variants of the SSIM index and NIQE that are mathematically amenable and lend themselves to being used as regularizers in the WGAN-GP optimization framework. Importantly, we empirically demonstrate the ability of our approach in augmenting the capability of the generator.

### 3.1 Quality Aware BWGAN Regularization

From the definitions in (3) and (7), the SSIM index is bounded in the interval [-1, 1] that immediately renders it an invalid distance metric (since it can take negative values). This makes the direct application of the SSIM index in the flexible BWGAN framework infeasible. Brunet et al. [BVW11] have analyzed the mathematical properties of SSIM index and show that valid distance metrics can be derived from the components of the SSIM index defined in (4). For e.g., $d^1(P_{(i,j)}, T_{(i,j)})) := \sqrt{1 - L(P_{(i,j)}, T_{(i,j)})}$, $d^2(P_{(i,j)}, T_{(i,j)})) := \sqrt{1 - CS(P_{(i,j)}, T_{(i,j)})}$ are shown to be valid normalized distance metrics. Importantly, they show that

$$d^Q(P_{(i,j)}, T_{(i,j)}) = \sqrt{2 - L(P_{(i,j)}, T_{(i,j)}) - CS(P_{(i,j)}, T_{(i,j)})} \tag{10}$$

is a valid distance metric that also preserves the quality discerning properties of the SSIM index. Further, as in the SSIM index, the image level distance metric for an $M \times N$ image is defined as

$$d^Q(P, T) = \frac{1}{M \times N} \sum_{i=1}^{M} \sum_{j=1}^{N} d^Q(P_{(i,j)}, T_{(i,j)}). \tag{11}$$

We refer the reader to [BVW11] for a detailed exposition of the properties of this distance metric. We call this a quality aware distance metric that serves as a good candidate for regularizing GANs. We hypothesize that by making the discriminator Lipschitz with respect to $d^Q(X, Y)$ in the image space,

the gradients computed from such a regularized discriminator emphasize the structural information in the generated images. Further, since $d^Q(X, Y)$ is bounded below and we are operating in the general metric space of images, we are in a position to impose the Lipschitz constraint directly. In order to do so, we follow the approach suggested in BWGAN [AL18] and introduce a gradient penalty regularization term of the form

$$\text{SSIM GP} = E_{X \sim P_r, Y \sim P_G} \left[ \left( \frac{|D(X) - D(Y)|}{d^Q(X, Y)} \right) - 1 \right]^2. \tag{12}$$

In addition to the SSIM GP regularizer, we also employ a 1-GP regularizer (i.e., WGAN-GP) to ensure stable training. The overall discriminator loss function is

$$L_d = \min_{D \epsilon \mathcal{D}} \left( E_{z \sim P_z} D(G(z)) - E_{X \sim P_r} D(X) \right) + \lambda_1 E_{\hat{x} \sim P_{\hat{x}}} (||\nabla_{\hat{x}} D(\hat{x})||_2 - 1)^2 +$$

$$\lambda_2 E_{X \sim P_r, G(z) \sim P_G} \left[ \left( \frac{|D(X) - D(G(z))|}{d^Q(X, G(z))} \right) - 1 \right]^2, \tag{13}$$

where $\lambda_1$ and $\lambda_2$ are empirically chosen. Further, as in the WGAN-GP setting, $\hat{x}$ is sampled from a line joining the real and fake image distributions. Our coupled gradient penalty with respect to $d^Q(X, Y)$ imposes a strong Lipschitz constraint. We believe that this quality aware discriminator penalty results in a quality aware GAN formulation.

### 3.2 Quality Aware Gradient Penalty

As discussed in Section 2.2.2, the MSCN coefficient distribution of pristine and distorted images of natural scenes have a unique statistical signature. We claim that if the MSCN coefficient statistics possess a unique and reliably consistent signature, then so must the MSCN coefficient statistics of the spatial gradients and discriminator gradients of natural images. We empirically show that this is indeed the case, and that a NIQE-like formulation works well in quantifying the naturalness of discriminator gradients as well. Fig. 1b shows the empirical histograms of the MSCN spatial gradient image of a representative natural scene and its distorted versions. Fig. 1c shows the empirical histograms of the MSCN discriminator gradient image of a representative natural scene and its distorted versions. We have chosen a minimally trained ($\sim$ 100 iterations) deep neural network for finding the discriminator gradients. We make the following observations about the histograms: a) unimodal, b) Gaussian-like, c) distortions affect statistics. All these observations are identical to the MSCN coefficient statistics of natural images shown in Fig. 1a. We believe that the discriminator gradients show this statistical behavior since discriminators are smooth functions and natural images possess a unique local statistical signature. Another visual example that clearly illustrates our observations and motivates our work is shown in Fig. 2.

Based on these observations, we propose a NIQE-like score for the naturalness of the discriminator gradients of natural scenes. As in NIQE, a GGD is used to model the MSCN coefficients of pristine image discriminator gradients, and an AGGD is used to model the product of these coefficients. Finally, an MVG is used to model the parameters of the GGD and AGGD of MSCN of discriminator gradients from pristine images. The pristine discriminator gradient MVG model is characterized by its parameters $\mu_P, \Sigma_P$. These parameters represent the class of all discriminator gradients computed with respect to pristine natural images.

The naturalness of a test discriminator gradient image $T$ is computed to be its "distance" from the pristine image gradient class and is given by

$$||(T|\mu_P, \Sigma_P)||_{\text{NIQE}} := \sqrt{(\mu_P - \mu_T)^T \left( \frac{\Sigma_P + \Sigma_T}{2} \right)^{-1} (\mu_P - \mu_T)}, \tag{14}$$

where $\mu_T, \Sigma_T$ are the model parameters of the test image's MVG model. This function serves as a quality aware gradient penalty that we use to regularize a GAN as discussed next. As an aside, Fig. 1b shows that our hypothesis would work even if a spatial gradient regularizer is used. As discussed in Section 2, several regularization approaches have been proposed in the literature to improve the image quality, stability, generalization ability of GANs. These include regularizing with non-zero mean and zero mean gradient penalty, Sobolev norm penalty etc. While these approaches consider

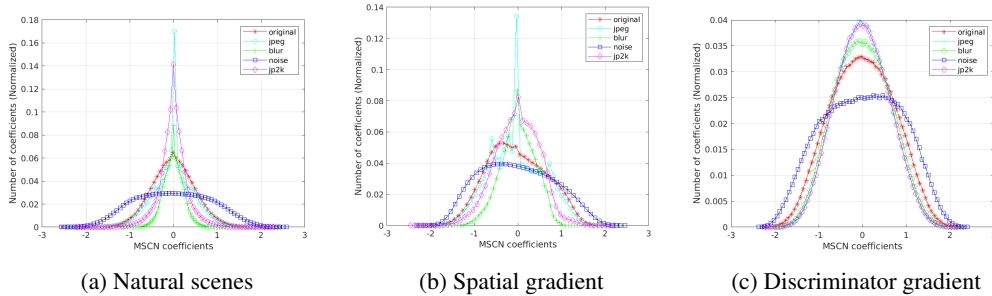

|  (a) Natural scenes | (b) Spatial gradient | (c) Discriminator gradient |

Figure 1: The empirical histograms of MSCN coefficients. (1a) Pristine natural scenes and their distorted versions. (1b) Spatial gradient of pristine natural scenes and their distorted versions. (1c) Discriminator gradient of pristine natural scenes and their distorted versions.

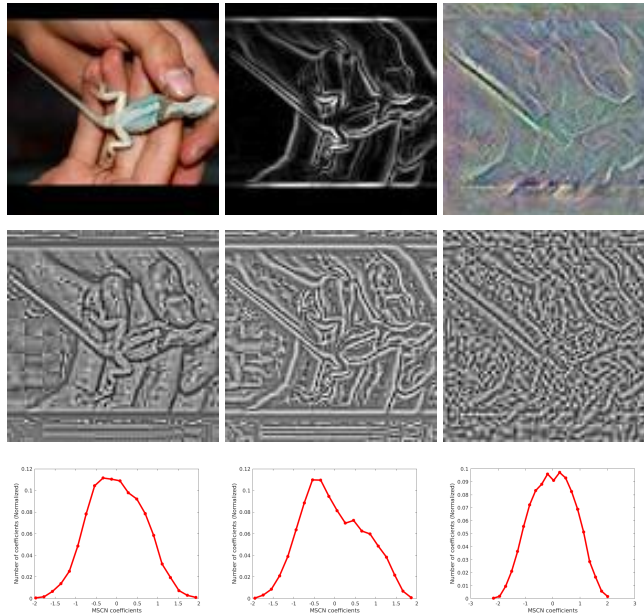

Figure 2: Top row (L-R) shows a real image, its corresponding spatial gradient and discriminator gradient maps. Middle row (L-R) shows their corresponding mean subtracted contrast normalized (MSCN) coefficients. Bottom row (L-R) shows the normalized histograms of the respective MSCN coefficients.

the norm of the discriminator gradient, they do not make use of the local correlation present in the gradient. Based on our hypothesis on the statistics of the discriminator gradient values of natural scenes presented in the previous section, we propose a novel regularization term that helps impose these statistical constraints on the generated images. We have shown through Fig. 1c that $\nabla_x D(x)$ contains useful local spatial statistics information. Our regularizer is designed to force the local statistics of the discriminator gradient of $\hat{x}$ to be as close to those of real images as possible. Our claim is that such a regularization strategy results in improving visual quality of the generated images. The NIQE "distance" function in (14) serves as the statistics preserving regularizer. As mentioned earlier, we work in the WGAN-GP framework to demonstrate our method. The overall discriminator cost function includes the 1-GP regularizer and the NIQE function regularizer as defined in

$$
L_d = \min_{D \epsilon \mathcal{D}} \left( E_{z \sim P_z} D(G(z)) - E_{X \sim P_r} D(X) \right) + \lambda_1 E_{\hat{x} \sim P_{\hat{x}}}(||\nabla_{\hat{x}} D(\hat{x})||_2 - 1)^2 +
$$
$$
\lambda_2 E_{\hat{x} \sim P_{\hat{x}}}(||(\nabla_{\hat{x}} D(\hat{x})|\mu_P, \Sigma_P)||_{\text{NIQE}}),
$$
(15)

where $\lambda_1$ and $\lambda_2$ are hyper parameters chosen empirically. As before, $\hat{x}$ is sampled from a line joining the real and fake image distributions.

## 4 Experiments

**Datasets:** We have evaluated the efficacy of proposed regularizers on three datasets: 1) CIFAR-10 [35] ($60K$ images of $32 \times 32$ resolution), 2) CelebA [Liu+15]($202.6K$ face images cropped and resized to resolution $64 \times 64$. 3) STL-10 [CNL11] ($100K$ images of resolution $96 \times 96$ and $48 \times 48$).
**Network Details:** All our experiments are done using a residual architecture for discriminator and generator used in WGAN-GP [Gul+17]. Batch normalization is applied to each resnet layer in the generator. We have used Adam as the optimizer with the standard momentum parameters $\beta_1 = 0$. and $\beta_2 = 0.9$. The initial learning rate was set to 0.0002 for CIFAR-10 and STL-10 datasets and 0.0001 for the CelebA dataset. The learning rate is decreased adaptively. We have empirically chosen the hyper parameters $\lambda_1$ and $\lambda_2$ to be 1 and 0.1 respectively. All our models are trained for $100K$ iterations with a batch size of 64. The discriminator is updated five times for every update of the generator.
**Evaluation:** We evaluated our method using two quantitative measures: 1) Inception Score (IS) [Sal+16] and 2) Frechet Inception Distance (FID) [Heu+17]. IS measures the sample quality and diversity by finding the entropy of the predicted labels. Higher IS indicates a better model. FID score measures the similarity between real and fake samples by fitting a multi variate Gaussian (MVG) model to the intermediate representation for the real and fake samples respectively. In case of FID, lower scores indicate a better model. We have used $50K$ randomly generated samples for computing the inception score and FID score. For comparison with previous models, we have computed the FID scores for CIFAR-10 and CelebA datasets using the official Tensorflow implementation, and for computing the FID scores of STL-10 dataset, we have used the Chainer implementation used by SNGAN [Miy+18]. IS and FID are computed five times for the best model and the mean and variance are reported.
**Results:** Our primary motivation in this work is to formulate quality aware loss functions for GANs primarily in the non-progressive setting. We present representative samples from the SSIM based QAGAN in Fig. 3 and the NIQE based QAGAN in Fig. 4. IS and FID are reported in Tables 1, 2, 3 and 4.

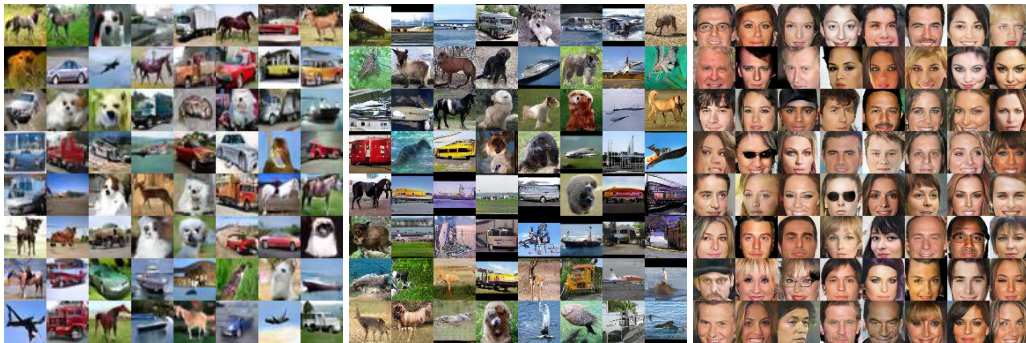

(a) CIFAR-10 dataset ($32 \times 32$).    (b) STL-10 dataset ($48 \times 48$).    (c) CelebA dataset ($64 \times 64$).

Figure 3: Randomly sampled images generated using QAGANs with quality aware distance metric regularizer (SSIM).

From the figures and tables, we see that QAGANs are very competitive with the state-of-the-art methods on all three datasets. Importantly and interestingly, QAGANs deliver consistently good performance with respect to both IS and FID, while other methods do well mostly with respect to IS. This provides clear quantitative evidence of the improved quality of images generated by QAGANs. Also, it underscores our claim that explicitly using objective IQA metrics in GAN cost functions is not only a promising way forward but also long overdue.

We see that the NIQE-based regularizer shows inconsistent performance only with respect to IS on the CIFAR-10 dataset. We attribute this inconsistency to the small image size ($32 \times 32$) of this dataset. This would lead to poorer estimates of the model parameters (compared to other resolutions) which in turn reduces performances.

Further, we believe that the proposed quality aware regularizers can be applied to progressive architectures as well. We demonstrate this by applying the proposed regularizers to the PGGAN

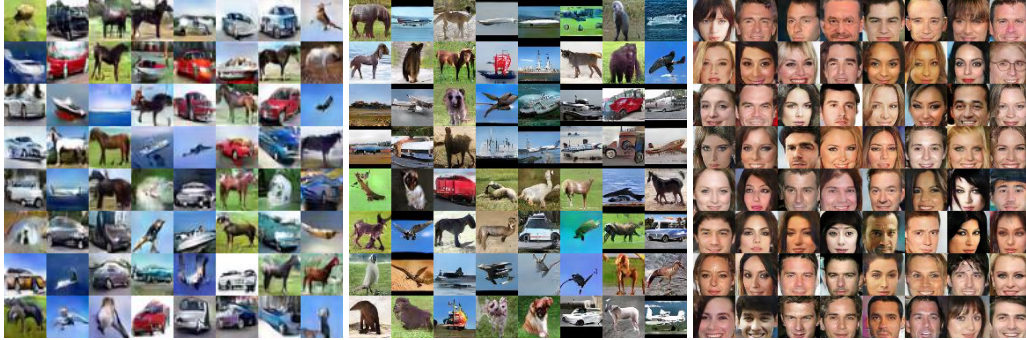

(a) CIFAR-10 dataset ($32 \times 32$).    (b) STL-10 dataset ($48 \times 48$).    (c) CelebA dataset ($64 \times 64$).

Figure 4: Randomly sampled images generated using QAGANs with quality aware gradient penalty regularizer (NIQE).

Table 1: Inception Score (IS) and Fréchet Inception Distance (FID) computed from 50,000 samples of the CIFAR-10 dataset ($32 \times 32$). Scores that are unavailable are marked with a '-'.

| Model | IS | FID |
|---|---|---|
| Real data | $11.24 \pm 0.12$ | 7.80 |
| DCGAN [RMC15] | $6.16 \pm 0.07$ | - |
| WGAN-GP [Gul+17] | $7.86 \pm 0.10$ | $40.2 \pm 0.0$ |
| CTGAN [Kim+18] | $8.12 \pm 0.12$ | - |
| SNGAN [Miy+18] | $8.12 \pm 0.12$ | $21.5 \pm 0.21$ |
| $W^{-\frac{3}{2},2}$ - Banach WGAN [AL18] | $8.26 \pm 0.07$ | - |
| $L^{10}$ - Banach WGAN [AL18] | $8.31 \pm 0.07$ | - |
| MMD GAN-rep-b [Li+17] | $8.29 \pm 0.0$ | $16.21 \pm 0,0$ |
| **QAGAN (SSIM)** | $\mathbf{8.37 \pm 0.04}$ | $\mathbf{13.91 \pm 0.105}$ |
| **QAGAN (NIQE)** | $\mathbf{7.87 \pm 0.027}$ | $\mathbf{12.4697 \pm 0.068}$ |

Table 2: FID on the CelebA dataset ($64 \times 64$).

| Model | FID |
|---|---|
| Real Faces (CelebA) | 1.09 |
| WGAN-GP [Gul+17] | 12.89 |
| Banach WGAN [AL18] | 10.5 |
| MMD GAN-rep-b [Li+17] | 6.79 |
| **QAGAN (SSIM)** | **6.421** |
| **QAGAN (NIQE)** | **6.504** |

Table 3: IS and FID on the STL-10 dataset ($48 \times 48$).

| Model | IS | FID |
|---|---|---|
| Real Data ($48 \times 48$) | $26.08 \pm 0.26$ | 7.9 |
| WGAN-GP [Gul+17] | $9.05 \pm 0.12$ | $55.1 \pm 0.0$ |
| SNGAN [Miy+18] | $9.10 \pm 0.04$ | $40.10 \pm 0.50$ |
| MMD GAN-rep [Li+17] | $9.36 \pm 0.0$ | $36.67 \pm 0.0$ |
| **QAGAN (SSIM)** | $\mathbf{9.29 \pm 0.05}$ | $\mathbf{19.77 \pm 0.0091}$ |
| **QAGAN (NIQE)** | $\mathbf{9.1720 \pm 0.08}$ | $\mathbf{19.45 \pm 0.0013}$ |

Table 4: IS and FID on the STL-10 dataset ($96 \times 96$).

| Model | IS | FID |
|---|---|---|
| **QAGAN (SSIM)** | $\mathbf{9.66 \pm 0.18}$ | $\mathbf{3.7155 \pm 0.004}$ |
| **QAGAN (NIQE)** | $\mathbf{8.948 \pm 0.01}$ | $\mathbf{3.1951 \pm 0.013}$ |

architecture [Kar+18] (both original and growing) at resolutions of $128 \times 128$ and $256 \times 256$ on the CelebA dataset. The qualitative results at an image resolution of $256 \times 256$ are shown in Fig. 5 and the quantitative results are given in Table 5. These results are shown after $6K$ iterations. Interestingly, we observed that the proposed regularizers resulted in faster convergence and improved visual quality of the generated images. The quantitative improvement in performance is clear from the FID values in Table 5.

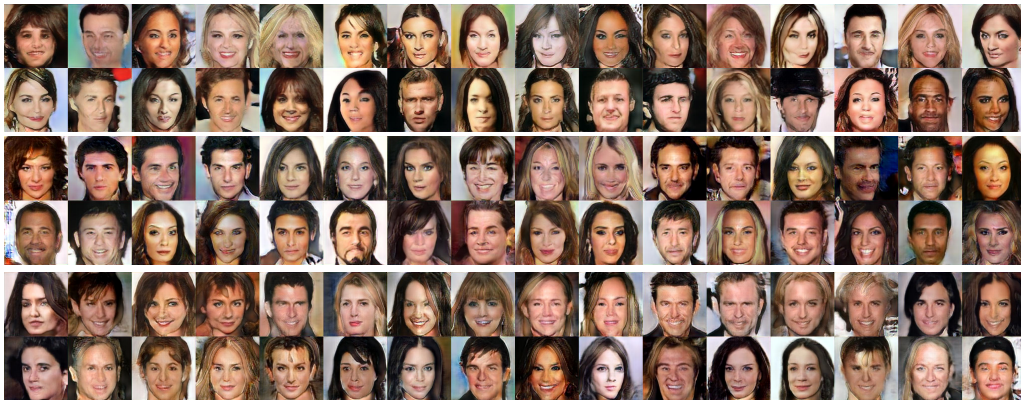

Figure 5: Randomly sampled images generated using QAGANs for CelebA dataset with a progressively growing architecture ($256 \times 256$) Top row: PGGAN [Kar+18]. Middle row: PGGAN with SSIM. Bottom row: PGGAN with NIQE.

Table 5: FID on the CelebA dataset for PGGAN.

| Model | FID |
|---|---|
| Resolution ($128 \times 128$) | |
| PGGAN [Kar+18] | 64.50 |
| **PGGAN + QAGAN (SSIM)** | **47.46** |
| **PGGAN + QAGAN (NIQE)** | **49.80** |
| Resolution ($256 \times 256$) | |
| PGGAN [Kar+18] | 62.86 |
| **PGGAN + QAGAN (SSIM)** | **40.83** |
| **PGGAN + QAGAN (NIQE)** | **47.27** |

# 5   Conclusions

Based on insights from both FR and NR IQA metrics, we have proposed two novel regularization approaches for the WGAN-GP framework. The key takeaway from our work is that the unique local structural and statistical signature of pristine natural images must be preserved in the generated images. We demonstrated how the SSIM and NIQE based regularizers guide the generator towards the class of pristine natural images and thereby ensure its unique local structural and statistical signature. The performance of QAGANs was shown to be very competitive with the state-of-the-art methods over three popular datasets. We believe that this work opens up new and exciting directions in image and video generative modeling, given the plethora of excellent QA metrics. The challenge however lies in translating the QA metrics into a form that fits the GAN framework.

# 6   Acknowledgement

We would like to thank Dr. J. Balasubramaniam from the Math department at IIT Hyderabad for his valuable suggestions and insights during this work. We would also like to thank NVIDIA for the GPU donation. SSC would like to acknowledge the sound track of *Kavaludaari* for inspiration while writing.

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
