[Supplementary Material · Supplementary_material_Quality_Aware_Generative_Models_916.pdf]

# Quality Aware Generative Adversarial Networks

We present supplementary material for Quality Aware Generative Adversarial Networks (QAGANs). The material includes the following:

1. Convergence analysis of QAGAN (SSIM)
2. Motivation for QAGAN (NIQE)
3. Convergence analysis of QAGAN (NIQE)
4. Overfitting analysis
5. Details of network architecture
6. Loss curves
7. More qualitative examples

## 1 Convergence analysis of QAGAN (SSIM)

Brunet et al. [BVW11] present a systematic analysis of the mathematical properties of the SSIM index. For completeness, we present a few results from their work that help in our convergence analysis. We follow the same notation and convention as in the paper draft. We start with two normalized metrics that they have defined and connected with the terms in the SSIM index.

**Normalized metrics:**

$$d^1(\mu_P, \mu_T) := \sqrt{\frac{||\mu_P - \mu_T||^2}{\mu_P^2 + \mu_T^2 + c_1}}$$

From this definition it follows that $L(P,T) = 1$ if and only if $\mu_P = \mu_T$ (where $L(P,T)$ is the image level mean term of the SSIM index). Also, $d^1(\mu_P, \mu_T)$ can be equivalently written as

$$d^1(\mu_P, \mu_T) := \sqrt{1 - L(P,T)}.$$

Similarly, $CS(P,T) = 1$ if and only if $P - \mu_P = T - \mu_T$ and $d^2(P,T) := \sqrt{1 - CS(P,T)}$ can be equivalently written as

$$d^2(P - \mu_P, T - \mu_T) := \sqrt{\frac{||(P - \mu_P) - (T - \mu_T)||^2}{||P - \mu_P||^2 + ||T - \mu_T||^2 + (N-1)c_2}},$$

$$d^2(P - \mu_P, T - \mu_T) := \sqrt{\frac{\sigma_P^2 - 2\sigma_{PT} + \sigma_T^2}{\sigma_P^2 + \sigma_T^2 + c_2}},$$

$$d^2(P - \mu_P, T - \mu_T) := \sqrt{1 - CS(P,T)}$$

where $CS(P,T)$ is a combination of the contrast and structure comparison terms. The similar forms of $d^1(\mu_P, \mu_T)$ and $d^2(P - \mu_P, T - \mu_T)$ can be written as the normalized root mean square error (NRMSE) as defined in the following:

$$d(x,y) = NRMSE(x,y,c) = \frac{||x - y||_2}{||x||_2 + ||y||_2 + c},$$

where $c$ is a stabilizing constant.

NRMSE was proved to be a metric in the vector space $R^N$ for $c \geq 0$. Then, a vector valued metric $d(P,T)$ is formulated from the two normalized metrics $d^1(\mu_P, \mu_T)$ and $d^2(P - \mu_P, T - \mu_T)$.

$$d(P,T) = (d^1(\mu_P, \mu_T), d^2((P - \mu_P), (T - \mu_T)))$$

taking the weighted $l^2$ norm of the metrics $d^1(\mu_P, \mu_T)$ and $d^2(P - \mu_P, T - \mu_T)$ with the weights $w = (1,1)$ and $p = 2$ leads to a metric.

$$d^Q(P,T) := ||d(P,T)||_2 := \sqrt{||d^1(\mu_P, \mu_T)||_2 + ||d^2((P - \mu_P), (T - \mu_T))||_2}$$

$$d^Q(P,T) := ||d(P,T)||_2 := \sqrt{2 - L(P,T) - CS(P,T)}$$

Based on this formulation, it is shown that the distance function $d^Q(P,T)$ derived from the components of the SSIM index is a valid distance metric between two images $P, T$.

**Theoretical and perceptual validation:** They have also verified that the distance metric $d^Q(P,T)$ is equivalent to the original SSIM index theoretically. The metric $d^Q(P,T)$ is the lower order estimation of the $\sqrt{1 - SSIM(P,T)}$ as shown below:

$$
\begin{aligned}
\sqrt{1 - SSIM(P,T)} &= \sqrt{1 - L(P,T)CS(P,T)} \\
&= \sqrt{1 - ((1 - d^1(\mu_P, \mu_T))^2)((1 - d^2((P - \mu_P), (T - \mu_T)))^2)} \\
&= \sqrt{(d^1(\mu_P, \mu_T))^2 + (d^2((P - \mu_P), (T - \mu_T)))^2 - (d^1(\mu_P, \mu_T))(d^2((P - \mu_P), (T - \mu_T)))}
\end{aligned}
$$

Further,

$$||d(P,T)||_2 \approx \sqrt{(d^1(\mu_P, \mu_T))^2 + (d^2_{PT}((P - \mu_P), (T - \mu_T)))^2)}$$

which can be equivalently written as $||d(P,T)||_2 \approx \sqrt{1 - SSIM(P,T)}$

The quality aware distance metric

$$d^Q(P,T) = \sqrt{2 - L(P,T) - CS(P,T)} \tag{1}$$

is a valid distance metric that preserves the quality discerning properties of the SSIM index. Also,

$$0 \leq d^Q(P,T) \leq \sqrt{2}. \tag{2}$$

When two images are equal (i.e., $P = T$ or perceptually identical), then $d^Q(P,T) = 0$. If two images are far apart perceptually then $d^Q(P,T) = \sqrt{2}$

To show that $d^Q(P,T)$ is a good approximation of $\sqrt{1 - SSIM(P,T)}$ with respect to its quality assessment capability, they conducted an experiment on the LIVE [SSB06] and TID-2008 [Pon+09] IQA databases with the realistic distorted images. First, they directly compared the two object metrics. The correlation between $d^Q(P,T)$ and $\sqrt{1 - SSIM(P,T)}$ was found to be 0.967. Further, they got correlations of 0.948 and 0.839 between $d^Q(P,T)$ and subjective scores on the LIVE and TID 2008 databases respectively.

**Convergence analysis of QAGAN (SSIM):**

- Mescheder et al. [MGN18] have proved in their analysis that WGAN-GP is not convergent since it does not have negative eigenvalues for its Jacobian matrix. They have also proved that unregularized and WGAN-GP methods will have energy solutions and work well in practice.

- As our gradient penalty term with respect to SSIM $d^Q(P,Q)$ also imposes a bound on the gradient and falls into the category of 1-GP, we can't prove the convergence property of QAGAN (SSIM). However, we claim that since we are operating on top of the WGAN-GP loss function, and from the boundedness property of $d^Q(P,T)$ in(2), our extra SSIM based gradient penalty term will not adversely effect the GAN training and will also have energy solutions.

- Empirically, our SSIM based regularization worked well and did not have any instability issues.

## 2 Motivation for QAGAN (NIQE)

We present more empirical evidence to substantiate our claim about the local statistical properties of spatial gradient maps and discriminator gradient maps $\nabla_x D(x)$.

We have taken inspiration from the Natural Image Quality Evaluator (NIQE) [MSB13] for introducing the discriminator gradient penalty regularizer in our quality aware approach. Before describing NIQE, we define how an image can be mean subtracted and contrast normalized (MSCN). Suppose $X$ is a natural image from the set of natural images. The MSCN coefficients of $X$ are obtained by subtracting the local mean from the image and then dividing with its local standard deviation.

$$\hat{X}(i,j) = \frac{X(i,j) - \mu(i,j)}{\sigma(i,j) + 1},$$

where $\mu(i,j)$ is the local mean of an $N \times N$ region centered around $(i,j)$ and $\sigma(i,j)$ is the standard deviation of the same region.

NIQE relies on the following observations about the statistics of naturals scenes: a) the statistics of MSCN natural images reliably follow a Gaussian distribution, b) the statistics of MSCN pristine and distorted images can be modeled well using a generalized Gaussian distribution (GGD).

We claim in our work that the unique local statistical signature of natural images carry over to spatial gradients and more importantly, to discriminator gradients. We make this claim about discriminator gradients based on the observation that even a minimally trained discriminator function is smooth. We provide empirical evidence to this claim in Figs. 1, 2 and 3. It is quite clear from these visual examples that both spatial gradients and discriminator gradients of real images do indeed have a unique statistical signature in them. Specifically, their MSCN coefficients are unimodal and lend themselves to being modeled by a GGD.

Thus, the motivation for our discriminator gradient penalty regularizer is to ensure that the generated images preserve this unique statistical property. In order to find the "distance" of the discriminator gradient map of a generated image from discriminator gradient map of the class of real (or pristine) images, we proposed a function whose form is identical to NIQE. Therefore, we present how NIQE computes this "distance."

- MSCN pristine image patch coefficients are modeled using a generalized Gaussian density (GGD), and the products of neighbouring MSCN coefficients are modeled using an asymmetric GGD (AGGD).

- The parameters of the GGD and AGGD are used to formulate the Natural Scene Statistic feature vectors from the image patches.

- NSS feature vectors are computed from image patches of a large corpus of pristine natural images.

- Let $x_1, x_2, \ldots, x_k$ denote the NSS feature vectors computed from the corpus of natural image patches,

- A multivariate Gaussian (MVG) model is fit to the NSS feature vectors of pristine image patches.

$$f_X(x_1, x_2, \ldots, x_k) \sim \frac{1}{2\pi |\Sigma_P|^{\frac{1}{2}}} \times \exp(-\frac{1}{2}(x - \mu_P)^T \Sigma_P (x - \mu_P)).$$

  The parameters of this fitted MVG distribution $(\mu_P, \Sigma_P)$ represent the class of pristine natural images.

The quality of a test image $T$ is measured in terms of the "distance" of its MVG parameters $\mu_T, \Sigma_T$ from the pristine MVG parameters. This is quantified as:

$$NIQE(T) = D(\mu_P, \mu_T, \Sigma_P, \Sigma_T) = \sqrt{(\mu_P - \mu_T)^T \left( \frac{\Sigma_P + \Sigma_T}{2} \right)^{-1} (\mu_P - \mu_T)}, \qquad (3)$$

assuming that the sum of the matrices is invertible. The NIQE score measures how far a given image is from the class of pristine natural scenes. Mittal et al. [MSB13] have reported a linear correlation score of 0.9147 between NIQE and subjective scores on the LIVE IQA database [SSB06].

(a) Real image        (b) Spatial gradient        (c) Discriminator gradient

(d) MSCN of real image     (e) MSCN of spatial gradient     (f) MSCN of discriminator gradient

(g) Real image        (h) Spatial gradient        (i) Discriminator gradient

Figure 1: Top row shows real image, its corresponding spatial gradient and discriminator gradient maps. Second row shows their corresponding mean subtracted contrast normalized (MSCN) coefficients. Bottom row shows the normalized histograms of MSCN maps.

In the proposed quality aware gradient penalty, we simply replace MSCN real images in the previous steps with MSCN discriminator gradient maps of real images.

We demonstrate the effectiveness of this regularizer in Fig. 4. We see that the histogram of the MSCN coefficients of the discriminator gradient map of generated images clearly approaches the histogram of discriminator gradient map of real images as the generator reaches optimality. We attribute this to the guidance provided by the quality aware gradient penalty.

## 3 Convergence Analysis of QAGAN (NIQE)

Based on the definition of the NIQE function in (12), we present a few of its properties in the following.

**Properties of NIQE:**

- $NIQE(X) \geq 0$.

(a) Real image      (b) Spatial gradient      (c) Discriminator gradient

(d) MSCN of real image    (e) MSCN of spatial gradient    (f) MSCN of discriminator gradient

(g) Real image      (h) Spatial gradient      (i) Discriminator gradient

Figure 2: Top row shows real image, its corresponding spatial gradient and discriminator gradient maps. Second row shows their corresponding mean subtracted contrast normalized (MSCN) coefficients. Bottom row shows the normalized histograms of MSCN coefficients.

- $NIQE(X)$ can be written as the Euclidean norm

$$NIQE(X) = \sqrt{(\mu_P - \mu_X)^T S_X (\mu_P - \mu_X)},$$

 where

$$S_X = \left( \frac{\Sigma_P + \Sigma_X}{2} \right)^{-1}.$$

- Since NIQE assumes that $\Sigma_P + \Sigma_T$ is invertible, $S_X$ is a positive definite matrix. By spectral decomposition, there exists a matrix $U_X$ such that $S_X = U_X^T U_X$. Therefore,

$$NIQE(X) = \sqrt{(\mu_P - \mu_X)^T U_X^T U_X (\mu_P - \mu_X)}.$$

- Thus, $NIQE(X)$ can be written as

$$NIQE(X) = ||U_X \mu_{PX}||_2 \tag{4}$$

 where $U_X$ is the spectral decomposition matrix of $S_X$ and $\mu_{PX} = \mu_P - \mu_X$. We make use of these properties in our convergence proof.

Figure 3: Real images and their corresponding discriminator gradient maps.

Figure 4: Demonstration of the effectiveness of the proposed quality aware gradient penalty.

**Convergence proof:** In order to prove the convergence of our regularizer, we use the approach proposed by Mescheder et al. [MGN18]. They have proved that adding the extra regularizers (zero centered penalties make the GAN training stable by computing the eigen values of the Jacobian matrix.)

Following the same procedure, we prove that our NIQE based quality aware gradient penalty also makes the GAN training stable and convergent.

While we prove this for the standard GAN setting, it applies to the WGAN framework as well. Suppose the generator $G$ is parameterized $\theta$ and the discriminator $D$ is parameterized by $\psi$. The objective function can be described as follows:

$$L(\theta, \psi) = E_{z \sim p(z)}[f(D_\psi(G_\theta(z)))] + E_{x \sim p_D(x)}[f(-D_\psi(x))] \tag{5}$$

The choice of the function $f$ decides the variant of the GAN. For the DCGAN objective function: $f(t) = -\log(1 + \exp(-t))$ while for WGAN objective function: $f = t$.

$f$ has to be continuously differentiable and $f'(t) = 0$ for all $t \in R$. The goal of the GAN training is to minimize the objective function in (5) with respect to generator parameters $\theta$ and to maximize the objective function with respect to discriminator parameters $\psi$.

Let the parameters at Nash equilibrium be $(\theta^*, \psi^*)$. The gradient vector field $v(\theta, \psi)$ is given by

$$v(\theta, \psi) = \begin{pmatrix} -\nabla_\theta L(\theta, \psi) \\ \nabla_\psi L(\theta, \psi) \end{pmatrix}. \tag{6}$$

Mescheder et al. [MGN18] and Nagarajan et al. [NK17] have shown that local convergence can be analyzed by the eigenvalues of the Jacobian $v'(\theta, \psi)$ at stationary point $(\theta^*, \psi^*)$. The Jacobian $v'(\theta^*, \psi^*)$ is given by the following for the unregularized case

$$v'(\theta^*, \psi^*) = \begin{bmatrix} -\nabla_\theta^2 L(\theta^*, \psi^*) & -\nabla_{\theta,\psi}^2 L(\theta^*, \psi^*) \\ \nabla_{\theta,\psi}^2 L(\theta^*, \psi^*) & \nabla_\psi^2 L(\theta^*, \psi^*) \end{bmatrix} \tag{7}$$

This can be equivalently written as the following by taking into the assumptions made by Mescheder et al. [2].

$$v'(\theta^*, \psi^*) = \begin{bmatrix} 0 & -K_{DG}^T \\ K_{DG} & K_{DD} \end{bmatrix}, \tag{8}$$

where $K_{DD}$ and $K_{DG}$ matrices are defined as follows

$$K_{DD} = 2f''(0)E_{p_D(x)}[\nabla_\psi D_{\psi^*}(x)\nabla_\psi D_{\psi^*}(x)^T],$$

$$K_{DG} = f'(0)\nabla_\theta E_{p_\theta}[\nabla_\psi D_{\psi^*}(x)]|_{\theta=\theta^*}.$$

For the regularization case, the gradient vector field and Jacobian matrices are the following.

$$v(\theta, \psi) = \begin{pmatrix} -\nabla_\theta L(\theta, \psi) \\ \nabla_\psi L(\theta, \psi) - \nabla_\psi R_q(\theta, \psi) \end{pmatrix}. \tag{9}$$

$$v'(\theta^*, \psi^*) = \begin{bmatrix} -\nabla_\theta^2 L(\theta^*, \psi^*) & -\nabla_{\theta,\psi}^2 L(\theta^*, \psi^*) \\ \nabla_{\theta,\psi}^2 L(\theta^*, \psi^*) & \nabla_\psi^2 L(\theta^*, \psi^*) - \nabla_\psi^2 R_q(\theta, \psi) \end{bmatrix} \tag{10}$$

This can be equivalently written as the following by taking into the assumptions made by Mescheder et al. [MGN18].

$$v'(\theta^*, \psi^*) = \begin{bmatrix} 0 & -K_{DG}^T \\ K_{DG} & K_{DD} - L_{DD} \end{bmatrix} \tag{11}$$

Here our proposed penalty aim is to minimize $NIQE(\nabla_{\hat{x}}(D(\hat{x})))$. We operate only on the discriminator and so it is only a function of parameters $\psi$.

$$R_q(\psi) = \frac{\gamma}{2}E_{p(\hat{x})}(NIQE(\nabla_{\hat{x}}(D_\psi(\hat{x})))),$$

where $\gamma$ is the hyper parameter, which is equal to $2\lambda_2$. By taking the directional derivative $\nabla_{\hat{x}}(D_\psi(\hat{x}))$ as the test image, NIQE score will be estimated by the following

$$D(\mu_P, \mu_T, \Sigma_P, \Sigma_T) = \sqrt{(\mu_P - \mu_T)^T \left[\frac{\Sigma_P + \Sigma_T}{2}\right]^{-1} (\mu_P - \mu_T)}, \tag{12}$$

The parameters $\mu_{T,\psi}$ and $\Sigma_{T,\psi}$ are the MVG parameters obtained from the directional derivative $\nabla_{\hat{x}}(D_\psi(\hat{x}))$. So the $\mu_{T,\psi}$ and $\Sigma_{T,\psi}$ are also the function of parameters of discriminator ($\psi$). Let us consider matrix $S_{PT,\psi}$ and vector $\mu_{PT,\psi}$

$$S_{PT,\psi} = \left[\frac{\Sigma_P + \Sigma_{T,\psi}}{2}\right]^{-1}, \tag{13}$$

$$\mu_{PT,\psi} = (\mu_P - \mu_{T,\psi}), \tag{14}$$

$$NIQE(\nabla_{\hat{x}}(D_\psi(\hat{x}))) = \sqrt{\mu_{PT,\psi}^T S_{PT,\psi} \mu_{PT,\psi}}. \tag{15}$$

As in NIQE, we assume that the matrix $S_{PT,\psi}$ is positive definite. Then by spectral theorem for symmetric matrix. There exist a diagonal matrix $(\Lambda_\psi) = \text{diag}(\lambda_1, \lambda_2, \lambda_3, \dots, \lambda_n)$ and an orthogonal matrix $Q_\psi$ such that $Q_\psi^T = Q_\psi^{-1}$ and

$$S_{PT,\psi} = Q_\psi^T \Lambda_\psi Q_\psi.$$

Let the matrix $S_{PT,\psi}$ be a positive definite matrix. $\lambda_1 \geq 0, \lambda_2 \geq 0, \lambda_3 \geq 0, \ldots, \lambda_n \geq 0$.

$$U_\psi = \text{diag}(\sqrt{\lambda_1}, \sqrt{\lambda_2}, \sqrt{\lambda_3}, \sqrt{\lambda_4}, \ldots, \sqrt{\lambda_n})Q_\psi,$$

Note that,

$$S_{PT,\psi} = U_\psi^T U_\psi,$$

and $\mu_{\hat{PT},\psi} = U_\psi.\mu_{PT,\psi}$.

Equivalently, $NIQE(\nabla_{\hat{x}}(D_\psi(\hat{x})))$ can be written as the Euclidean norm $||\mu_{\hat{PT},\psi}||_2$.

$$NIQE(\nabla_{\hat{x}}(D_\psi(x))) = \sqrt{\mu_{PT,\psi}^T S_{PT,\psi}\mu_{PT,\psi}}, = ||\mu_{\hat{PT},\psi}||_2.$$

Our NIQE based regularizer will be written as the following

$$R_q(\theta,\psi) = \frac{\gamma}{2}E_{p(\hat{x})}[||U_\psi.\mu_{\hat{PT},\psi}||^2]$$

the derivatives for the regularizer $R_q(\psi)$ with respect to $\psi$ are derived as follows.

$$\nabla_\psi R_q(\theta,\psi) = \gamma E_{p(\hat{x}}[\nabla_\psi U_\psi.\mu_{\hat{PT},\psi}U_\psi.\mu_{\hat{PT},\psi}],$$

The second derivative $\nabla_\psi^2 R_q(\psi)$ with respect to parameters $\psi$ at the optimal point $(\theta^*, \psi^*)$ are derived as follows

$$\nabla_\psi^2 R_q(\theta^*,\psi^*) = \gamma E_{p_{\hat{x}}}[(\nabla_\psi U_{\psi^*}.\mu_{\hat{PT},\psi^*})(\nabla_\psi U_{\psi^*}.\mu_{\hat{PT},\psi^*})^T]$$

$$L_{DD} = \nabla_\psi^2 R_q(\theta^*,\psi^*)$$

Our quality aware regularizer based on NIQE introduces the new term $L_{DD}$ term into the Jacobian matrix. Other than this, all terms in the Jacobian matrix remain the same.

$$v^T L_{DD} v = \gamma E_{p(\hat{x})}(||\nabla_\psi U_{\psi^*}\mu_{\hat{PT},\psi^*}||^2)$$

$$v^T L_{DD} v \geq 0$$

$K_{DD}$ remains the same as in Mescheder et. al [2]. Therefore, we can conclude that

$$v^T K_{DD} v \leq 0$$

and it follows that the matrix $M_{DD} = K_{DD} - L_{DD}$ will be a negative definite matrix.

By this, we have proved that the Quality Aware Gradient Penalty regularizer $R_q$ has not affected the properties of the Jacobian matrix. It is convergent.

## 4 Overfitting

Figure 5 shows the histograms of the discriminator's weights for the WGAN-GP, QAGAN (SSIM) and QAGAN (NIQE) approaches respectively. We observe that our proposed regularizers make the discriminator network more sparse compared to WGAN-GP. The weight distributions of WGAN-GP, QAGAN (SSIM) and QAGAN (NIQE) are as follows: $[-1.73551, 1.4799]$, $[-0.9153, 1.0258]$ and $[-.051438, 0.42510]$ respectively. Variances of the weights with WGAN-GP, QAGAN (SSIM) and QAGAN (NIQE) are $0.03804, 0.007948$ and $0.002483$ respectively. Clearly, the proposed quality aware regularizers avoid overfitting.

## 5 Loss curves

Figure 6 shows that discriminator loss with WGAN-GP saturates quickly, while our QAGAN (SSIM) consistently increases. This shows that the objective function provided by QAGAN (SSIM) gives more informative gradients to the generator. Figure 7 shows that the discriminator loss with QAGAN (NIQE) also increases consistently.

Histogram of the weights of the discriminator

Figure 5: Weight distribution of discriminator block for various methods.

Table 1: Generative Model Architecture for CIFAR-10 data set

| Discriminator $D(x)$ | Generator $G(z)$ |
|---|---|
| Input image :$(3\times32 \times 32)$ | Input latent vector size :128 |
| Residual block, Resample=Down :$(128 \times16 \times 16)$ | Linear-dense layer : :$(128 \times4 \times 4)$ |
| Residual block, Resample=Down:$(128 \times8 \times 8)$ | Residual block, Resample=Up:$(128 \times8 \times 8)$ |
| Residual block, Resample=-:$(128 \times8 \times 8)$ | Residual block, Resample=Up:$(128 \times16 \times 16)$ |
| Residual block, Resample=-:$(128 \times8 \times 8)$ | Residual block, Resample=Up:$(128 \times32 \times 32)$ |
| ReLU- Mean pooling: 128 | Convolutional block,:$(3 \times32 \times 32)$ |
| Linear : 1 | Tanh :$(3 \times 32 \times 32)$ |

Table 2: Generative Model Architecture for STL-10 dataset

| Discriminator $D(x)$ | Generator $G(z)$ |
|---|---|
| Input image :$(3\times48 \times 48)$ | Input latent vector size :128 |
| Residual block, Resample=Down :$(128 \times24 \times 24)$ | Linear-dense layer : :$(128 \times6 \times 6)$ |
| Residual block, Resample=Down:$(128 \times12 \times 12)$ | Residual block, Resample=Up:$(128 \times12 \times 12)$ |
| Residual block, Resample=Down:$(128 \times6 \times 6)$ | Residual block, Resample=Up:$(128 \times24 \times 24)$ |
| Residual block, Resample=-:$(128 \times6 \times 6)$ | Residual block, Resample=Up:$(128 \times48 \times 48)$ |
| ReLU-Mean pooling: 128 | Convolutional block,:$(3 \times48 \times 48)$ |
| Linear : 1 | Tanh :$(3 \times 48 \times 48)$ |

Figure 6: Convergence curves of discriminator loss for WGAN-GP and QAGAN (SSIM).

## 6   Network Architecture

We have used the ResNet architecture for discriminator and generator and is a faithful reconstruction of the architecture proposed in the WGAN-GP framework [Gul+17]. We have used ReLU non-linearity. The input to the generator is of dimension 128. Batch normalization was used in the generator. The kernel size used in the all convolutional and residual layers is $[3 \times 3 \times 2]$.

For CelebA dataset ($64 \times 64$), We have used the same CIFAR-10 ResNet architecture reported in Table 1. We have added the up sampling residual block at the end in the generator to get to $64 \times 64$ and have added the extra down sampling residual block in the discriminator. For STL-10 data set of resolution $48 \times 48$, we have used a latent vector of size 128. The latent vector is mapped to $6 \times 6$. We have used the network architecture mentioned in Table 2. For STL-10 data set of resolution $96 \times 96$, we have added up/down sampling layers to generator and discriminator respectively.

## 7   Representative Samples

In Figs. 7-14, we present more samples from both the proposed Quality Aware GANs for all the data sets: CIFAR-10, STL-10 and CelebA. We also show the generated samples from STL-10 ($96 \times 96$) data set. Our claim that QAGANs' emphasis on the local structural information and local statistical information leads to improved image quality is evident from the qualitative samples below.

Through this work, our primary goal is to demonstrate that statistics of natural scenes, and the ability of IQA algorithms to quantify the naturalness of a scene have an important role to play in the generative modeling of natural images. The WGAN-GP framework provides us a good setting to

Figure 7: Convergence curves of discriminator loss for various methods.

convey our idea effectively. Nevertheless, we believe that the core idea of imposing a "naturalness" constraint in generating natural scenes would be effective wherever the discriminator function is smooth. This includes the 1-Lipschitz functions in WGAN-GP and PGGAN, hinge-loss objective function based Self-attention GAN and Spectral GAN, CT-GAN and WGAN-LP etc. Also, since we impose no constraints on the generator, we expect it to work well in the conditional GAN setting too.

## 8  Experiments with Progressively Growing architecture (PGGAN) [Kar+18]

To further justify our claim, we have applied the proposed regularizers to the PGGAN architecture (both original and growing) at resolutions of $128 \times 128$ and $256 \times 256$ on the CelebA dataset, and show the results in Fig. 17. The original PGGAN framework uses WGAN-GP [Gul+17] loss function.Interestingly and importantly, we observed that the proposed regularizers resulted in faster convergence and improved visual quality of the generated images. We hope that these results also address concerns about the effectiveness of QAGANs at higher resolutions. While memory and time constraints limited our testing to a resolution of $256 \times 256$ and 6K iterations, we are optimistic that our method would work at higher resolutions as well.

Figure 8: Randomly sampled generated images for CelebA dataset for Quality Aware Distance Metric.

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

(a) .

(b) .

(c) .

Figure 10: Randomly sampled generated images for STL-10 dataset for Quality Aware Distance Metric.($48 \times 48$)

Figure 11: Randomly sampled generated images fro STL-10 dataset for Quality Aware Distance Metric.(96 × 96)

Figure 12: Randomly sampled generated images for CelebA dataset for Quality Aware Gradient Penalty.

Figure 13: Randomly sampled generated images for CIFAR-10 dataset for Quality Aware Gradient Penalty.

Figure 14: Randomly sampled generated images for STL-10 dataset for Quality Aware Gradient Penalty.($48 \times 48$)

Figure 15: Randomly sampled generated images fro STL-10 dataset for Quality Aware Gradient Penalty.($96 \times 96$)

Figure 16: Resolution: $128 \times 128$. Top: PGGAN, $FID_{128\times128}$ = 64.50. Middle: PGGAN with SSIM, $FID_{128\times128}$ = 47.46. Bottom: PGGAN with NIQE, $FID_{128\times128}$ = 49.80 These are results after 6K iterations on the CelebA dataset.

Figure 17: Resolution: $256 \times 256$. Top: $FID_{256 \times 256}$ = 62.86. Center: PGGAN with SSIM, $FID_{256 \times 256}$ = 40.834. Bottom: PGGAN with NIQE, $FID_{256 \times 256}$ = 47.27. These are results after 6K iterations on the CelebA dataset.