[Reviews · NeurIPS 2019]

Reviewer 1



Update after the rebuttal: I stand by my review and rating. The additional experiments and explanations in the rebuttal largely clarify the concerns I had. --- The paper proposes an approach to improving training of Generative Adversarial Networks (GANs) on images. The idea is to use regularizers based on image-specific similarity metrics (SSIM, NIQE). The method is evaluated on non-progressive GANs trained on three datasets: CIFAR-10, STL-10, CelebA. The proposed method seems to substantially improve FID and IS relative to baselines. Pros: 1) Reasonable and, to my knowledge, new idea. 2) Quite clear and complete presentation 3) The results are good, both qualitatively and quantitatively, especially in terms of FID. 4) Extensive supplementary material with many details and extra results. Cons: 1) Not a huge technical innovation, but a rather incremental modification based on existing techniques. 2) Some questionable statements: - The usage of SSIM as loss function has been limited (Section 3). It definitely has been used in many works, one example is [1] below. - Boundedness to [-1, 1] immediately renders SSIM an invalid distance metric (sec. 3.1) - why? - Why does d^Q serve as a good candidate for regularizing GANs? [1] Loss Functions for Image Restoration with Neural Networks, Hang Zhao, Orazio Gallo, Iuri Frosio, Jan Kautz. Transactions on Computational Imaging, 2017 Overall, the paper proposes a reasonable approach, presents it well and shows that the method performs well empirically. I think the paper can be published.

Reviewer 2



Thanks for the rebuttal. I have read it carefully. The new experiments look good, but the authors do not seem to respond to my concern over SSIM metric between unpaired images. I keep my original review and rating. ----------------------------------------------------------- I think the paper is clear, the intuition is well claimed. Given all the prior works that smooth GAN training, the idea that integrates image quality assessment metrics with GANs sounds interesting. From the experiment samples, it seems that the quality aware gan does improve the sample quality, the generated CelebA and STL images look sharp. I would like to see the results of combining QAGAN with large scale GANs such as PGGAN or BigGAN. I think the semantic details and structures play a center role in the quality of large images. Quantitatively, the paper shows that QAGAN is able to achieves comparable IS and lower FID than baselines. Below are some comments. - sec 3.1 and 3.2 discuss how to integrate SSIM into GAN framework, sec 3.3 and 3.4 discuss how to combine NIQE with GAN. I suggest the authors put sec 3.1 and 3.2 together, same for sec 3.3 and sec 3.4. - The MSCN coefficients for images generated by GAN in the supplementary material is useful, I suggest putting it in the main paper with figure 1. - I think this paper [1] is related to the main idea and should be compared and discussed in the paper in detail. - SSIM is orignally used to compare paired distorted image and its pristine counterpart. However, in QAGAN, the distance is calculated between randomly sampled generated image and real image, which are not paired. I am not sure whether the SSIM distance still makes sense in this setting. Chances are that the distance between two real images can be even larger than the distance between real image and distorted image because the two real images may have completely different local luminance, contrast and structure. [1] Kancharla, P., & Channappayya, S. S. (2018, October). Improving the Visual Quality of Generative Adversarial Network (GAN)-Generated Images Using the Multi-Scale Structural Similarity Index. In 2018 25th IEEE International Conference on Image Processing (ICIP) (pp. 3908-3912). IEEE.

Reviewer 3



Update: I thank the authors for their feedback. The additional experiments address my concerns regarding larger scale experiments and the discussion about the choice of lambdas clarifies my concerns regarding convergence. Thus, I decided to raise my score. ######################################################################################################################################## (1) Summary of the Paper This work advocates the usage of image quality objectives as regularizers when training GANs in order to obtain more natural looking images. In particular, two different metrics are used: (i) a variant of the SSIM index and (ii) a gradient penalty inspired by NIQE along the WGAN GP framework in order to achieve a superior performance to several baselines, as demonstrated empirically on 3 datasets. (2) Paper Clarity The writing of the paper is clear and it is easy to follow. It is well motivated and contains enough background information. (3) Methodology and Significance The idea to use image quality metrics as regularizers is simple, yet seems intuitive, which I appreciate. However, while the new regularizers are shown to have a superior performance in terms of FID/INC, I have several concerns/questions. Specifically: - How do the regularizers affect the stability of training? How sensitive are they to the choice of the hyperparameters (the lambdas)? - In Table 1 and 2, are the scores provided for FID and INC computed using the same trained model (i.e. same lambdas), or are they tuned separately for both scores? Overall, while the method is simple and intuitive and it appears to work well, its contribution is somewhat limited. The main reason behind this is that it is only applied to one particular GAN variant i.e. WGAN GP and it is not clear how one can extend it to other GAN versions. Furthermore, since the main motivation of the submission is to generate more natural looking images, a proper evaluation should include some higher-resolution datasets where the regularizers can potentially have a bigger impact.

[Author Response · NeurIPS 2019]

We thank the reviewers for their careful review and insightful comments. We address the comments in the following.

**Limited nature of the contribution:**

Through this work, our primary goal is to demonstrate that statistics of natural scenes, and the ability of IQA algorithms to quantify the naturalness of a scene have an important role to play in the generative modeling of natural images. The WGAN-GP framework provides us a good setting to convey our idea effectively. Nevertheless, we believe that the core idea of imposing a "naturalness" constraint in generating natural scenes would be effective wherever the discriminator function is smooth. This includes the 1-Lipschitz functions in WGAN-GP and PGGAN, hinge-loss objective function based Self-attention GAN and Spectrally normalized GAN, CT-GAN and WGAN-LP etc. Also, since we impose no constraints on the generator, we expect it to work well in the conditional GAN setting too.

To further justify our claim and implement reviewer suggestions, we have applied the proposed regularizers to the PGGAN architecture (both original and growing) at resolutions of $128 \times 128$ and $256 \times 256$ on the CelebA dataset, and show the results in Fig. 1. Interestingly and importantly, we observed that the proposed regularizers resulted in faster convergence and improved visual quality of the generated images. We hope that these results also address concerns about the effectiveness of QAGANs at higher resolutions. While memory and time constraints limited our testing to a resolution of $256 \times 256$ and 6K iterations, we are optimistic that our method would work at higher resolutions as well.

Figure 1: Top: $128 \times 128$. Bottom: $256 \times 256$. Left: PGGAN, $FID_{128 \times 128} = 64.50$, $FID_{256 \times 256} = 62.86$. Center: PGGAN with SSIM, $FID_{128 \times 128} = 47.46$, $FID_{256 \times 256} = 38.324$. Right: PGGAN with NIQE, $FID_{128 \times 128} = 49.80$, $FID_{256 \times 256} = 44.84$. These are results after 6K iterations on the CelebA dataset.

**Comparison with the work by Kancharla and Channappayya, ICIP 2018 [Kancharla2018]:**

While their work is similar in spirit, we present several fundamental differences in the following. First, our work clearly discusses the issues with the direct application of IQA algorithms as cost functions and proposes novel perceptual quality regularizers that are fine-tuned to the GAN framework - either that nicely fit the GAN math framework (SSIM-based) or that capture/model the local statistics of discriminator gradients (NIQE-based). Kancharla2018 on the other hand does a straightforward application of the MS-SSIM index and uses NIQE only for performance evaluation and not as a cost function. Next, our work presents a systematic stability analysis in the WGAN-GP setting and guarantees stability (please see supplementary material) while Kancharla2018 only presents empirical analysis. Also, they mention instabilities when the MS-SSIM term is given higher weightage. Further, we have conducted detailed experimental analysis and validation in our work. Finally, a qualitative comparison is shown in Fig. 2 from which it is clear that our method not only generates images with better structural information but also has greater diversity. This can be attributed to two main factors: our quality based regularizers and the improvements due to WGAN-GP relative to BEGAN.

Figure 2: Left image: Montage from Kancharla2018 (permission obtained from IEEE), with FID = 205. Right image: Montage from QAGAN with NIQE with FID = 86 (50K iterations). Note improved structural information and diversity.

**Clarifications:**

- We do not intend to portray that the SSIM index has not been used as a cost function in the literature. Rather, what we want to convey is that while IQA algorithms are indeed very effective, their usage as cost functions has not been as widespread as one would like due to their typically unwieldy mathematical formulation.

- Since the SSIM index can be negative, it no longer satisfies the requirement of a metric in the mathematical sense (i.e., $x, y \in \mathcal{X}$ for some set $\mathcal{X}, d(x, y) \geq 0$). We do not imply that the boundedness renders it an invalid metric.

- WGAN-GP uses the average of the error norm between the real and fake samples (without correspondence) as one of the elements of the cost function (Proposition 1, primal form in [Gul+17]). We reason that $d^Q$ would be a better choice than error norm (in the average sense) for measuring the *perceptual distance* between the real and fake image sets. We also observed that average $d^Q(X, Y)$ values reduce with iterations.

- We have presented a convergence/stability analysis (for any $\lambda > 0$) of the proposed regularizers in the supplementary material provided with the initial submission. We point the reviewer to Sections 1 and 3 in that document.

- The $\lambda$s were not tuned individually for FID and IS results. There were no stability issues with variation in $\lambda$s. Nevertheless, our choice of $\lambda$s is based on performance.

- We will incorporate presentation improvements in the final version if the submission is accepted.

Again, we thank the reviewers for their insightful comments that has led to important discussions and clarifications.

[Meta-Review · NeurIPS 2019]

The paper proposes a novel way to regularize training of deep adversarial generative models for natural images. The proposal is based on using the image quality metrics. While many different ways of stabilizing and regularizing GAN training were proposed in prior work, most of which based on various gradient penalties related to the Lipschitzness, this submission proposes an idea which is significantly different and novel. The paper evaluates the new method on three reasonably challenging datasets (CIFAR-10, STL-10, CelebA) and quantitatively shows objective advantages to other methods (in terms of FID and IS). The field of GANs and in particular various ways to stabilize their training has been recently attracting perhaps excessive amount of attention with many papers proposing multiple methods very similar in nature. Seeing sufficiently new ideas (even if they feel incremental) being introduced to this field is refreshing. I would support acceptance of this paper.